# γδ T cells are the prime antitumoral T cells in pediatric neuroblastoma

Suzanne M Castenmiller[1,2,3] , Anne L Borst[4], Leyma Wardak[1,2,3], Jan J Molenaar[4], Maria Papadopoulou[5,6,7,8] , Ronald R de Krijger[4,9], Alida FW van der Steeg[4], Max M van Noesel[4,10] , David Vermijlen[5,6,7,8], Rosa de Groot[1] , Judith Wienke[4] , Monika C Wolkers[1,2,3]

High-risk pediatric neuroblastoma patients have a dismal survival rate despite intensive treatment regimens. New treatment options are thus required. Even though HLA expression in neuroblastoma is low and immune cell infiltrates are limited, the presence of tumor-infiltrating lymphocytes (TILs) is indicative of better patient survival. Here, we show that most tumor lesions contain viable immune cell infiltrates after induction chemotherapy, with high percentages of CD3+ T cells. We therefore expanded the TILs and tested their antitumoral activity. With sufficient starting material, TIL expansion was as efficient as for adult solid tumors. However, whereas TIL products from adult tumors almost exclusively contained αβ T cells, in neuroblastoma-derived TIL products, γδ T cells expanded with similar efficacy as αβ T cells. Importantly, the antitumor responses in response to autologous tumor digest primarily originated from (Vδ1- and Vδ3-expressing) γδ T cells, and not from αβ T cells. In conclusion, this finding creates a window of opportunity for immunotherapy for neuroblastoma patients, with γδ T cells as potential prime responders.

## Introduction

Neuroblastoma is the most common extracranial solid cancer in children (Maris et al, 2007; Matthay et al, 2016). High-risk patients receive intensive treatment with debulking chemotherapy, surgical resection, and additional round(s) of high-dose chemotherapy followed by autologous stem cell rescue, radiotherapy, or the recently developed antibody-based anti-GD2 immunotherapy (Ladenstein et al, 2018). Yet, their 5-yr survival rate does not exceed

50% (Matthay et al, 2016; Li et al, 2023), highlighting the urgent need for novel treatment options.

Neuroblastoma is considered a "cold" tumor with low mutational tumor burden, low HLA-I expression, and low immune cell infiltrates (Pugh et al, 2013; Batchu, 2021; Wienke et al, 2021; Verhoeven et al, 2022). All this combined impedes the tumor cell recognition by T cells (Wölfl et al, 2004). However, encouraged by the effectiveness of the anti-GD2 immunotherapy in neuroblastoma patients (Yu et al, 2010; Theruvath et al, 2022; Flaadt et al, 2023), using the immune system to combat neuroblastoma has recently gained interest (Ladenstein et al, 2018; Zappa et al, 2023). Enhancing immune responses against neuroblastoma could indeed be powerful for improving the patients' prognosis.

Adoptive therapy with tumor-infiltrating lymphocytes (TIL therapy) is coming of age for treating solid tumors (Rosenberg & Restifo, 2015; Rohaan et al, 2022). Reinfusion of in vitro expanded autologous T cells from tumor lesions achieved high response rates in a phase 3 clinical trial in melanoma patients with disseminated tumors (Rohaan et al, 2022). Owing to this success and the recent FDA approval for TIL therapy for melanoma, TIL therapy is currently evaluated in other solid tumors, including non–small-cell lung cancer (Ben-Avi et al, 2018; De Groot et al, 2019; Creelan et al, 2021; Castenmiller et al, 2022), renal cell carcinoma (Asten et al, 2021), bladder cancer (Poch et al, 2018), and ovarian cancer (Pedersen et al, 2018). TIL therapy currently focuses on highly immune-infiltrated "hot" tumors with a high mutational burden. Yet, clinical responses were also reported for TIL therapy in so-called "cold" tumors with little to limited immune infiltrates and mutational burden, such as breast, ovarian, colorectal, and pancreatic cancers (Zacharakis et al, 2018, 2022; Kim et al, 2022; Amaria et al, 2024).

Even though neuroblastoma is considered a "cold" tumor, high TIL density positively correlated with the patient outcome (Batchu, 2021; Masih et al, 2021). Chemotherapy, the standard of care for

[1]Sanquin Blood Supply Foundation, Department of Research, T Cell Differentiation Lab, Amsterdam, The Netherlands    [2]Landsteiner Laboratory, Amsterdam Institute for Infection and Immunity, Cancer Center Amsterdam-Cancer Immunology, Amsterdam UMC, University of Amsterdam, Amsterdam, The Netherlands    [3]Oncode Institute, Utrecht, The Netherlands    [4]Princess Máxima Center for Pediatric Oncology, Utrecht, The Netherlands    [5]Department of Pharmacotherapy and Pharmaceutics, Université Libre de Bruxelles (ULB), Brussels, Belgium    [6]Institute for Medical Immunology (IMI), Université Libre de Bruxelles (ULB), Gosselies, Belgium    [7]ULB Center for Research in Immunology (U-CRI), Gosselies, Belgium    [8]WEL Research Institute, WELBIO Department, Wavre, Belgium    [9]Department of Pathology, University Medical Center Utrecht, Utrecht, The Netherlands    [10]Division Imaging and Cancer, University Medical Center Utrecht, Utrecht, The Netherlands

Correspondence: m.wolkers@sanquin.nl

high-risk neuroblastoma patients, promotes the influx of immune cells into tumors (Hong et al, 2011; Sistigu et al, 2014). We therefore hypothesized that neuroblastoma lesions after induction chemotherapy contain tumor-reactive T cells that could potentially be used for therapeutic purposes. Here, we report the effective generation of tumor-reactive TIL products from neuroblastoma tumor lesions. Importantly, the composition of pediatric TIL products substantially differed from that of TIL products from adult tumors. Indeed, $\gamma\delta$ T cells were enriched in neuroblastoma lesions compared with adult tumors, and this enrichment was maintained throughout the expansion protocol. The $\gamma\delta$ T cells expanded equally well as the $\alpha\beta$ T-cell receptor–expressing CD8[+] or CD4[+] T cells. Furthermore, $\gamma\delta$ T cells were the prime T-cell subset with antitumoral activity. Our findings thus uncover fundamental differences between adult and pediatric neuroblastoma–derived antitumor responses.

# Results

## T cells constitute most of the immune infiltrates in neuroblastoma lesions

We first studied the immune cell composition of neuroblastoma tumor lesions. We generated single-cell suspensions from 20 tumor lesions obtained from 18 patients after debulking surgery (Table S1). On average, 32.7% of the cells were viable, of which a median of 28.3% ± 23.0% were CD45[+] immune cells (Figs 1A–C and S1A). Owing to the cryopreservation of tumor digests before analysis, myeloid cells such as monocytes and neutrophils were underrepresented (Fig S1A). Of the remaining immune cells, CD3[+] T cells were with a median of 64.5% most abundant (Fig 1D). Comparative analysis with immunohistochemistry for three samples showed that the percentages obtained with flow cytometry provided a good representation of CD3[+] T-cell infiltrates (Fig S1D). Monocytes, B cells, NKT cells, and NK cells were also detected, but with highly variable percentages ranging from 0 to 40.2% (Figs 1D, S1A, and S2B).

CD3[+] T-cell infiltrates in neuroblastoma lesions primarily consisted of $\alpha\beta$ CD8[+] T cells and conventional $\alpha\beta$ CD4[+] T cells (Tconv), together accounting for a median of 80% of CD3[+] T cells (Figs 1E and S1A). CD127[low]CD25[+]Foxp3[+] regulatory T cells (Tregs) were detected in 17 out of 20 (85%) tumor lesions, yet with low percentages (Figs 1E and S1A). We also detected $\gamma\delta$ T cell infiltrates with a median of 9.2%, comprising up to 36% of the CD3[+] T cells in one tumor lesion (Figs 1E and S1A). Of note, the percentage of $\gamma\delta$ T cells of CD3[+] T cells reflects the percentages of $\gamma\delta$ T cells reported in tissues of children (Gray et al, 2024). The high variability in CD45[+] cell infiltrates and its composition could not be attributed to different disease stages (LR, low risk; MR, medium risk; HR, high risk) or different treatment regimens (Table S1). This finding contrasts significantly with the dogma that neuroblastomas are immunologically "cold" tumors, because almost all tumors contained immune infiltrates upon debulking chemotherapy, and the infiltrates primarily consist of T cells.

## TILs can be effectively expanded from neuroblastoma lesions

Having established that T cells are present in neuroblastoma lesions, we tested their capacity to expand. A recent study reported successful TIL expansion from neuroblastoma lesions (Hurtado et al, 2019); however, limited tumor reactivity had been observed (Hurtado et al, 2019). Because $\alpha$CD3/$\alpha$CD28 activation was used immediately for T cell expansion (Hurtado et al, 2019), this protocol may have favored the expansion of T cells from contaminating blood, or of non–tumor-reactive, bystander tissue-resident T cells that are amply present in solid tumors (Simoni et al, 2018; Li et al, 2022). This in turn may hamper the outgrowth of tumor-reactive, yet to some degree dysfunctional, T cells. We therefore used the rapid expansion protocol (REP [Besser et al, 2010; Dudley et al, 2010]), which for the first 2 wk of culture (pre-REP) uses only the addition of recombinant human IL-2 for TIL expansion from tumor digests (Berg et al, 2020; Rohaan et al, 2022). This culture setup allows for specific antigen presentation from the tumor cells in the pre-REP phase and is considered to support the survival and expansion of tumor-specific T cells, which should express the IL-2 receptor (Fig S2A) (Berg et al, 2020; Rohaan et al, 2022).

TILs from neuroblastoma lesions expanded on average 15-fold during the pre-REP phase, and 200-fold during the second 2 wk of expansion with $\alpha$CD3 and IL-2 (REP phase), resulting in a total expansion of ~1,500-fold (Fig 2A). However, the efficiency of TIL expansion was not uniform. Whereas 9 out of 20 (45%) TIL cultures expanded less than 500-fold (Fig 2A), 11 out of 20 (55%) TIL cultures expanded about 3,000-fold (Fig 2A), which is comparable to the TIL expansion rate reported for adult tumors (De Groot et al, 2019; Castenmiller et al, 2022; Rohaan et al, 2022). When we compared the patient's age (in months) at the time of tumor resection, we found a slight association with the efficiency of TILs to expand (Fig 2B, left panel). Treatment intensity negatively correlated with TIL expansion (Fig 2B, middle panel). Most prominently, the number of cells that were available for starting the TIL cultures significantly correlated with TIL expansion (Fig 2B, right panel). Low cell numbers primarily stemmed from tumor needle biopsies as input material (Table S1). In sum, we conclude that sufficient starting material from tumor lesions is pivotal to achieve efficient TIL expansion.

We next phenotyped the TIL products after REP culture (Fig S1B and E). Cell viability of the 19 measured TIL products was on average 65.84%, yet with high variability. Well-expanding TIL products (>500-fold) reached a median of 82.5% viability on average, which is above the threshold for clinical use (Fig 2C, black dots; Fig S1B). Poorly expanded TIL products (<500-fold) contained low numbers of CD45[+] cells (0–12.5%). In contrast, well-expanding TIL products contained 64.0–91.7% CD45[+] cells, and in nine samples, >90% of the CD45[+] cells were CD3[+] T cells (Fig 2D and E, black dots; Fig S1B). Only two TIL products with <100-fold expansion contained high percentages of viable cells (Fig S1F, *colored dots*), of which one contained almost exclusively CD45[+]CD3[+]CD8[+] T cells (Fig S1F, *blue dot*). CD19[+] B cells and Foxp3[+]CD4[+] T cells were absent in well-expanding TIL products, and only 2 out of 10 well-expanding TIL products contained low but detectable CD11c[+] or CD16[+] cells (Figs 2E and S1B). Thus, well-expanding TIL products contain high numbers of immune cells, of which the majority consists of T cells.

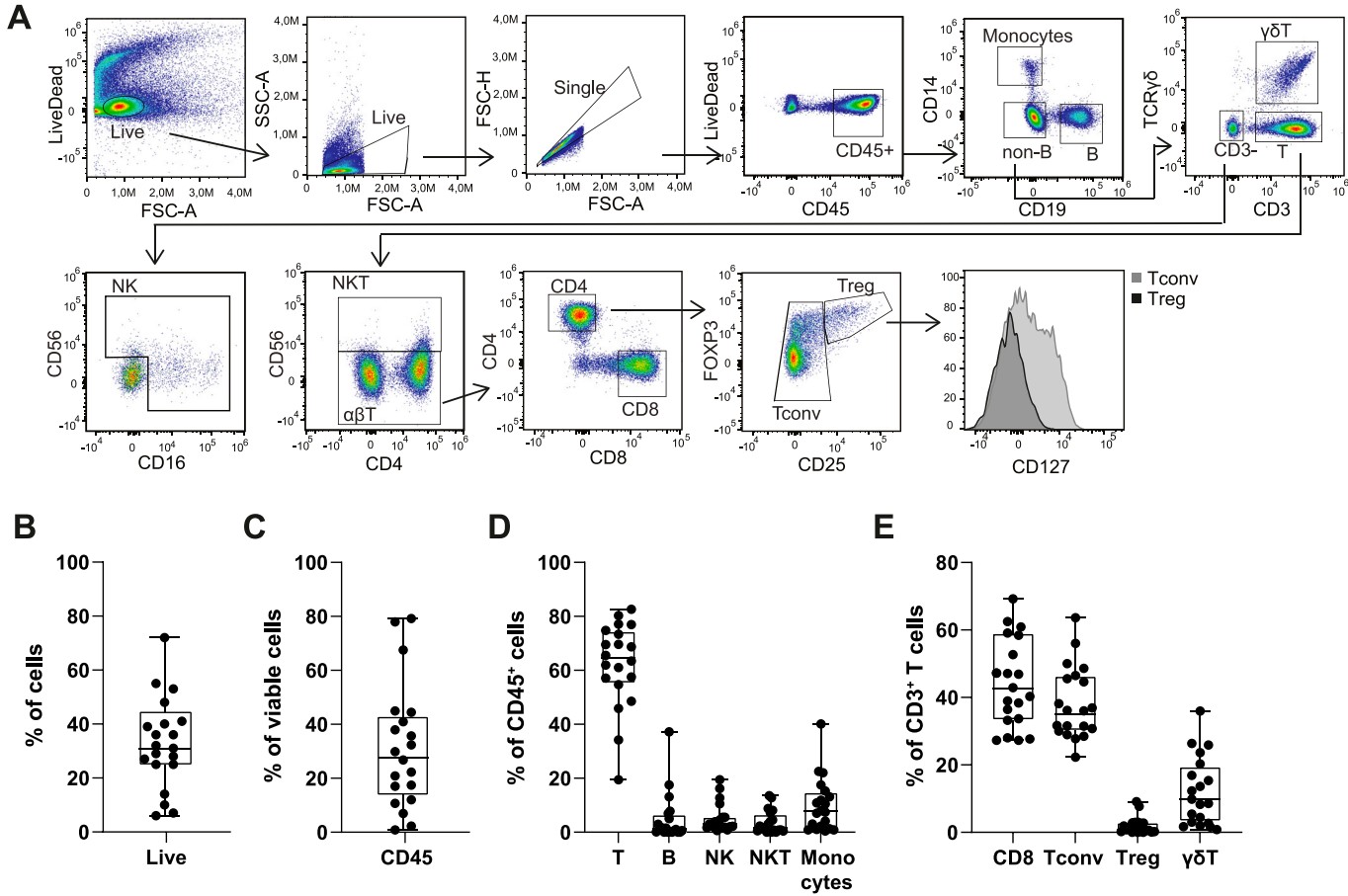

**Figure 1. Immune cell composition of pediatric neuroblastoma tumor lesions.**
**(A)** Representative gating strategy for analyzing immune cell infiltrates (sample M189AAF_2). **(B, C, D, E)** Percentage of (B) viable cells in neuroblastoma tumor lesions, (C) CD45⁺ immune infiltrates of viable cells, (D) different immune cell subsets as the percentage of total viable immune infiltrates, and (E) T-cell subsets as the percentage of total T-cell infiltrate. n = 20 samples; each dot represents one tumor sample. Box-and-whisker plots depict median, minimum, and maximum values (whiskers) and 25th and 75th pct (box).

Interestingly, when we compared the composition of CD3⁺ T-cell subsets in the tumor digest ex vivo with that of the expanded TIL products, we observed that the distribution between CD8⁺ T cells, Tconv cells, and γδ T cells was similar (Figs 1E and 2F; Table S1), indicating that the three T cell subsets expanded in a similar fashion (Fig 2G). We next compared the TIL composition from the neuroblastoma lesions with TILs from adult solid tumors (melanoma, NSCLC, RCC; Fig S2C). Even though adult TILs also contained γδ T cell infiltrates ex vivo, as previously reported (Girard et al, 2019; Wu et al, 2019, 2022), their percentages were lower than in pediatric neuroblastoma lesions (Fig 2H). Furthermore, γδ T cells derived from adult tumor lesions failed to expand (Fig 2H and I). Thus, only γδ T cells from pediatric neuroblastoma expanded well with the REP. In sum, TIL products can be generated from neuroblastoma lesions, yet its efficiency depends on the quantity of the starting material. In addition, the generated TIL products display a similar composition of effector T cells measured ex vivo.

## Expanded TILs from neuroblastoma display childhood-specific cytokine expression patterns

To test the functionality of TILs that were expanded from neuroblastoma lesions (Fig S1C), we focused on the well-expanded TIL products. All CD8⁺ TILs, Tconv TILs, and γδ TILs were potent producers of TNF after a 7-h stimulation with PMA/ionomycin (Fig S3A and B; see Fig 3A for an unstained control). However, the production of IFNγ was highly variable for all three T cell subsets (Fig S3A and B). Most strikingly, the production of IL-2 upon PMA/ionomycin stimulation was below 20% for CD8⁺ T cells and γδ T cells, and only four TIL products contained >20% IL-2–producing Tconv cells (Fig S3A and B). This limited IL-2 production corroborates with previous reports from blood-derived pediatric T cells (Schmiedeberg et al, 2016; Thome et al, 2016). Thus, neuroblastoma-derived TIL products can produce cytokines, but do so with lower potency than TIL products generated from adult solid tumors, which show very high production of all three cytokines with PMA/ionomycin stimulation (De Groot et al, 2019; Asten et al, 2021; Castenmiller et al, 2022).

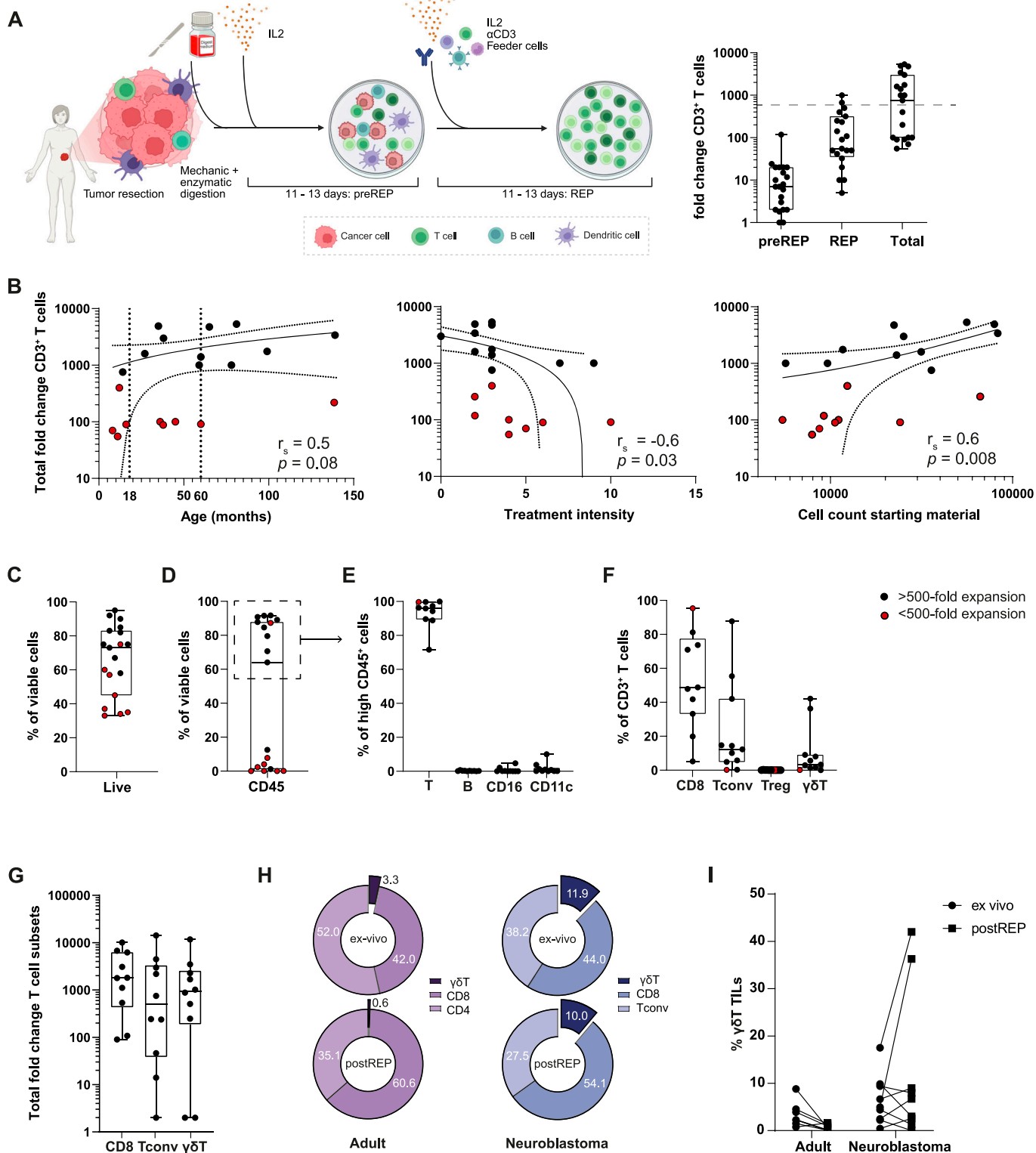

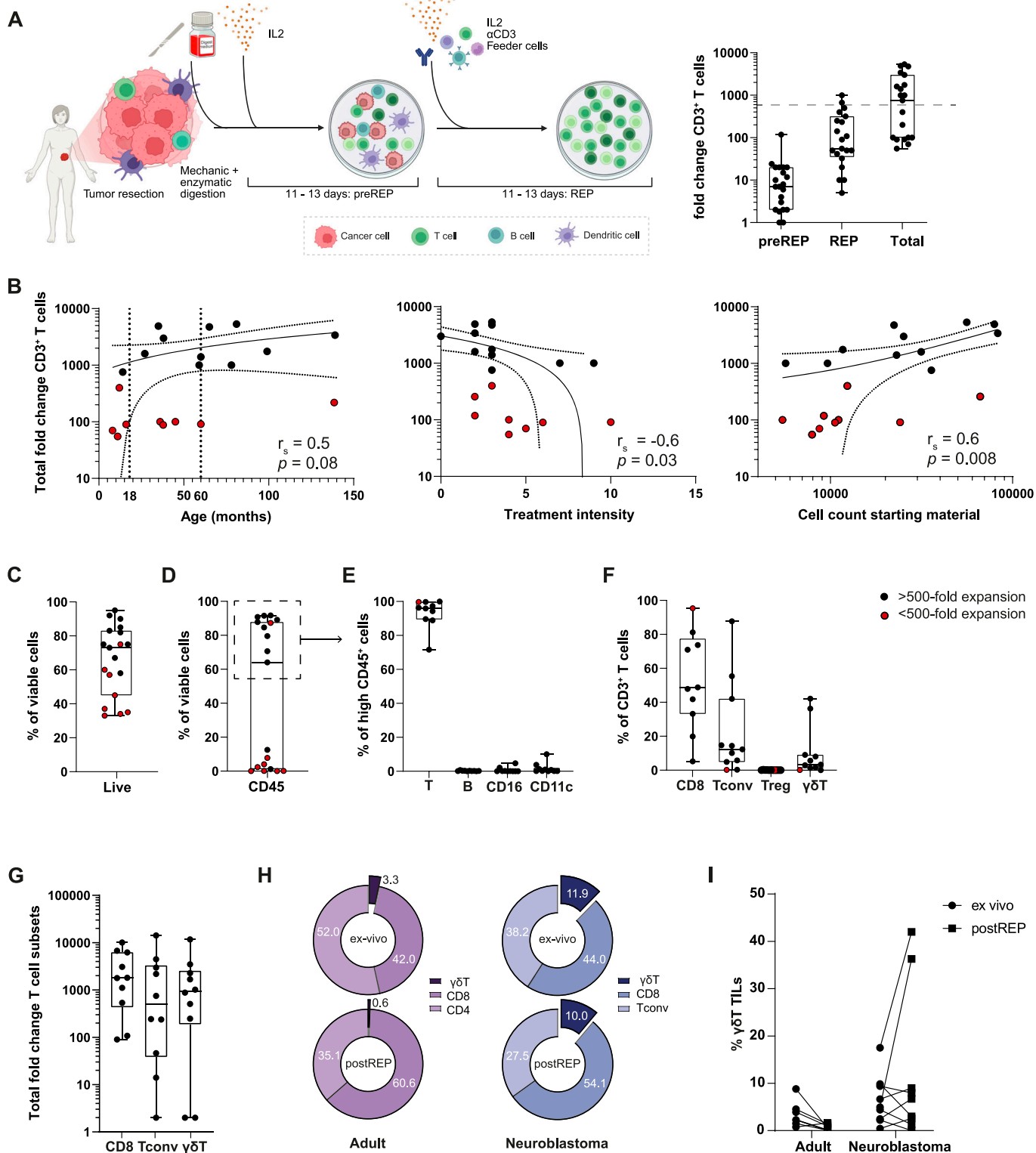

**Figure 2. Expansion potential and composition of tumor-infiltrating lymphocyte (TIL) products from neuroblastoma lesions.**
**(A)** Left: schematic overview of the expansion protocol. Right: fold expansion (right) of TILs during the first 10–13 d of the pre-rapid expansion phase (pre-REP; left), during the second 10–13 d (REP; middle), and pre-REP and REP combined (Total; right). The dotted line indicates 500-fold expansion, n = 20. **(B)** Correlation plots between total expansion (y-axis) and patients' age in months at the time of tissue collection (left), treatment intensity based on the total number of treatments received before surgery (middle), and total cell count of the starting material for TIL expansion (right). $r_s$ and adjusted *P*-values are indicated in each plot. *P*-values were corrected for six multiple comparisons using the Bonferroni method. Red dots indicate expansion below 500-fold (n = 9). **(C)** Percentage of viable cells in TIL products (n = 20). **(D)** Percentage of CD45[+] cells in TIL products (n = 19; one sample was excluded because of insufficient expansion). **(E, F)** Percentage of indicated immune cell subsets as the

## Tumor reactivity from neuroblastoma TIL products primarily originates from γδ T cells

We next tested whether the well-expanded TIL products contained tumor-reactive T cells. We exposed the TIL products for 6–7 h to autologous tumor digest, and we measured the expression of the activation marker CD137 (4-1BB) indicative of TCR triggering (Ye et al, 2014), the degranulation marker CD107a, and TNF and IFNγ required for effective antitumoral responses (Patel et al, 2017) (Fig 3A). To increase the low HLA expression cells of neuroblastoma cells (Wölfl et al, 2004), and thus their capacity to present antigens to T cells, we preexposed the tumor digests overnight with 1,000 IU/ml human recombinant (hr)IFNγ, before coculture with the TILs.

Coculture of TILs with autologous tumor digest resulted in significantly increased CD137 expression in all T-cell subsets when compared to medium control (Fig 3B; CD8+ T cells: P = 0.02; CD4+ T cells: P = 0.05; γδ T cells: P = 0.02). This finding indicates that the TCR was engaged. In contrast, CD8+ T cells and Tconv cells displayed very limited expression of CD107a or production of IFNγ, if present at all (Fig 3B). TNF production was also nearly absent in CD8+ T cells and limited, yet significantly increased, in Tconv cells (Fig 3B; P = 0.03). In contrast, γδ TILs not only significantly increased CD107a expression when exposed to tumor digest (Fig 3B; P = 0.008), but also substantially increased the production of TNF and IFNγ (Fig 3B; TNF: P = 0.03; IFNγ: P = 0.008). Of note, the observed antitumor response of γδ TILs did not depend on pretreating tumor digests with hrIFNγ (Fig S3C), suggesting that γδ TILs do not react to IFNγ response genes. Importantly, the observed γδ T-cell responses from neuroblastoma-derived TIL products were tumor-specific, as indicated by the significantly higher expression of CD137 (P = 0.02) and a trend in increased CD107a expression when compared to responses to autologous non-tumorous tissue digest (Fig 3C and D).

PD-1 and CD137 are two expression markers reported to enrich for tumor-reactive T cells (Patel et al, 2017; Thommen et al, 2018). Both markers were expressed on a subset of γδ T cells ex vivo (Fig S3D). The PD-1 expression was increased on conventional CD4 T cells, indicating that this T-cell subset also contained some tumor-reactive cells (Fig S3D). Nevertheless, the CD137 expression was significantly higher on γδ T cells than on αβ T cells (Fig S3D). However, neither the percentage of PD-1 nor CD137 ex vivo correlated with tumor reactivity of γδ T cells in the TIL product (Fig S3E). When we correlated the response rates of tumor-reactive γδ TILs (e.g., any γδ TIL that up-regulated at least one of the four measured markers), we observed a positive trend with the patient's age, but not with the percentage of γδ T cells ex vivo (Fig S3E). Thus, γδ T cells are the most tumor-reactive T-cell subset in neuroblastoma lesions.

## Vδ1 and Vδ3 γδ T cells are the prime source of antitumor responses

We next studied which γδ T-cell subset contributed most to the antitumor response. γδ T cells can be divided into subclasses based on the Vδ gene used in their TCR (Davey et al, 2017; Ravens et al, 2017; Papadopoulou et al, 2020b). In humans, Vδ1-, Vδ2-, and Vδ3-expressing γδ T cells are the most common subtypes (Papadopoulou et al, 2020b; Hu et al, 2023). We first measured the γδ T-cell composition of four expanded TILs with known antitumoral responses (Fig 4A). The Vδ1 subtype was most prevalent in three TIL products, and one primarily contained Vδ2 cells (Fig 4B). The percentage of Vδ3 cells was low but present in all four TIL products (Fig 4B). All γδ T-cell subsets were equally equipped to produce TNF and IFNγ when stimulated with PMA/ionomycin (Fig 4C and D; see Fig 4E for an unstained control). However, when we repeated the coculture with autologous tumor digests, increased CD137 expression was primarily found on Vδ1 cells and Vδ3 cells, and much less so on Vδ2 cells (Fig 4E and F). Increased CD107a expression was found on all γδ T-cell subsets but was most prominent on Vδ3 cells (Fig 4F). Cytokine production was also primarily detected on the Vδ1 and Vδ3 cells (Fig 4F). Thus, Vδ1 and Vδ3 cells are the prime γδ T-cell subsets responding to autologous tumor digests.

## Discussion

We here provide first evidence that TIL products can be generated from pediatric neuroblastoma tumor lesions with the TIL expansion protocol that is used to generate TIL products for melanoma patients (De Groot et al, 2019; Castenmiller et al, 2022; Rohaan et al, 2022). For practical reasons, we used frozen material for TIL expansion. Because the amount of viable cells was the best predictor for effective TIL expansion, and because fresh tumor material generally contains more viable cells, we hypothesize that using fresh material would possibly result in higher efficacy of generating tumor-reactive TIL products.

Importantly, we report that γδ T cells are the prime antitumoral T cells in these TIL products, which show higher response rates to the autologous tumor than to non-tumorous tissue digest. Whereas tumor-responsive CD8+ T cells in the TIL products failed to produce cytokines, CD4+ T cells not only displayed increased PD1 expression ex vivo (Marco et al, 2017) but also produced TNF in response to autologous tumor digest. This finding suggests CD4+ T cells could potentially also contribute to antitumoral T-cell responses in neuroblastoma, similar to what was previously reported for several adult solid tumors (Oja et al, 2017; Oh et al, 2020; Cachot et al, 2021; Nicolet et al, 2021; Kruse et al, 2023).

percentage of CD45+ cells, and (F) T cell subsets as the percentage of CD3+ cells. (G) Total fold expansion for CD8+, Tconv, and γδ T cell subsets. (E, F, G) n = 10; only TIL products with high percentages of CD45+ cells were included. (H) Indicated T-cell subsets as the percentage of CD3+ T cells ex vivo (n = 20) and after REP (n = 10). (I) γδ T-cell content as the percentage of CD3+ T cells ex vivo (dots) and upon REP (squares) for adult tumor samples (melanoma, n = 2; non–small-cell lung cancer, n = 3; renal cell carcinoma, n = 3) and for neuroblastoma (n = 10). Each dot represents one sample. Box-and-whisker plots depict median, minimum, and maximum values (whiskers) and 25th and 75th pct (box). Correlation plots depict the simple linear regression line with the 95th pct coincidence interval (dotted line).

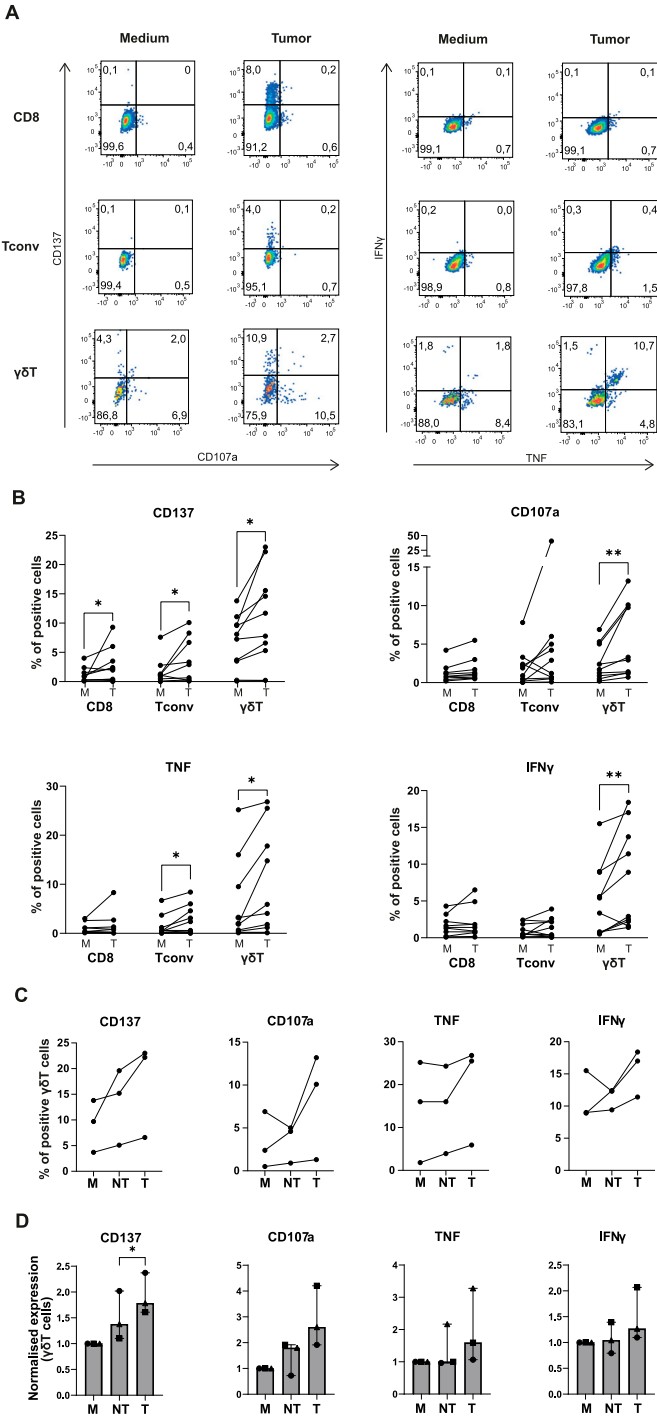

**Figure 3. γδ T cells display antitumor reactivity against autologous tumor digest.**
**(A)** Representative dot plots for expression levels of surface molecules (left, sample M959AAB) and cytokines (right, sample M583AAB) upon 6- to 7-h culture with medium alone, or with autologous tumor digest in a 1:1 ratio. **(B)** Compiled data of CD137, CD107a, TNF, and IFNγ expression on indicated T-cell subsets after culture with medium (M) or with autologous tumor digest (T). Each dot represents one tumor-infiltrating lymphocyte product (n = 10). Wilcoxon's test; * < 0.05, ** < 0.01. **(C)** Compiled data of CD137, CD107a, TNF, and IFNγ expression of γδ tumor-infiltrating lymphocytes after a 6- to 7-h coculture with medium (M), with non-tumorous tissue digest (NT), or with autologous tumor digest in a 1:1 ratio (T). **(D)** Data of (C) normalized to medium control. Each symbol indicates a different patient sample (n = 3). Tukey's multiple comparison test; * < 0.05.

For γδ T cells, it is yet to be determined how they recognize neuroblastoma cells. The γδ TCR does not have the obligate HLA restriction as the conventional αβ TCR (Vermijlen et al, 2018), and γδ T cells can kill target cells upon engagement of NK cell receptors (Silva-Santos & Strid, 2018). Whether the activation markers PD-1 and CD137 can be used to identify tumor-reactive T cells in the same fashion for γδ T cells as they do for αβ T cells is not fully understood. Yet, CD137 was recently used to detect virus-responding γδ T cells (Pei et al, 2020; Ji et al, 2024), and may therefore also be valid to identify tumor-reactive γδ T cells. Nevertheless, a more detailed analysis of γδ T cells to define the different γδ T-cell subtypes, and to profile checkpoint and activation markers, as well as NK cell receptors, is warranted.

Evidence that γδ T cells are contributors to antitumoral responses in solid tumors is recently accumulating. γδ T cells are predictive for overall increased survival in non–small-cell lung cancer and triple-negative breast cancer patients (Wu et al, 2019, 2022). Furthermore, γδ T cells respond to checkpoint inhibition and contribute to effective treatment responses in HLA-deficient colon cancer and in melanoma patients with a low mutational rate (de Vries et al, 2023; Davies et al, 2024). Neuroblastoma is also a tumor with a low mutational rate and low HLA expression. Because the antitumoral γδ T-cell responses did not substantially change when tumors were pretreated with hrIFNγ, it is conceivable that the tumor recognition of γδ T cells occurs independent of HLA. γδ T cells responded stronger to autologous tumor than to medium control, similar to what we previously observed for TILs from adult solid tumors (Fig 3C, [De Groot et al, 2019; Asten et al, 2021; Castenmiller et al, 2022]). However, because the expanded TILs respond more to autologous tumor digest than to fibroblasts, we conclude that these CD137-expressing, cytokine-producing γδ T cells are tumor-specific.

Importantly, the composition and responsiveness of TIL products from neuroblastoma substantially differs from that of TIL products generated from adult tumors. γδ TILs expand equally well as the αβ T-cell subsets, a feature that was not found in three types of adult solid tumors. This could be related to the inherent higher expansion capacity of γδ T cells early after birth (Papadopoulou et al, 2020a; Ravens et al, 2020; Giannoni et al, 2024). In neonates and young children, γδ T cells are more diverse (Davey et al, 2017). As children age, γδ T cells become more oligoclonal, decrease in quantity, and lose their capacity to expand (De Rosa et al, 2004; Schatorjé et al, 2012; Clark & Thomas, 2020). In addition, extrinsic factors could potentially alter the expansion capacity of pediatric γδ T cells, such as other immunostimulatory or immunosuppressive cells originating from the tumor digest during TIL expansion. The tumor microenvironment may again differ between pediatric and adult tumors, and even more so in these tumors with low mutational rates and limited inflammatory signature, as is the case for pediatric neuroblastoma.

Previous studies reported that adult γδ T cells may require other cytokines or stimuli in addition to IL-2 for sufficient expansion (Almeida et al, 2016; Verkerk et al, 2024). Even though we cannot exclude that additional stimuli may even further boost the

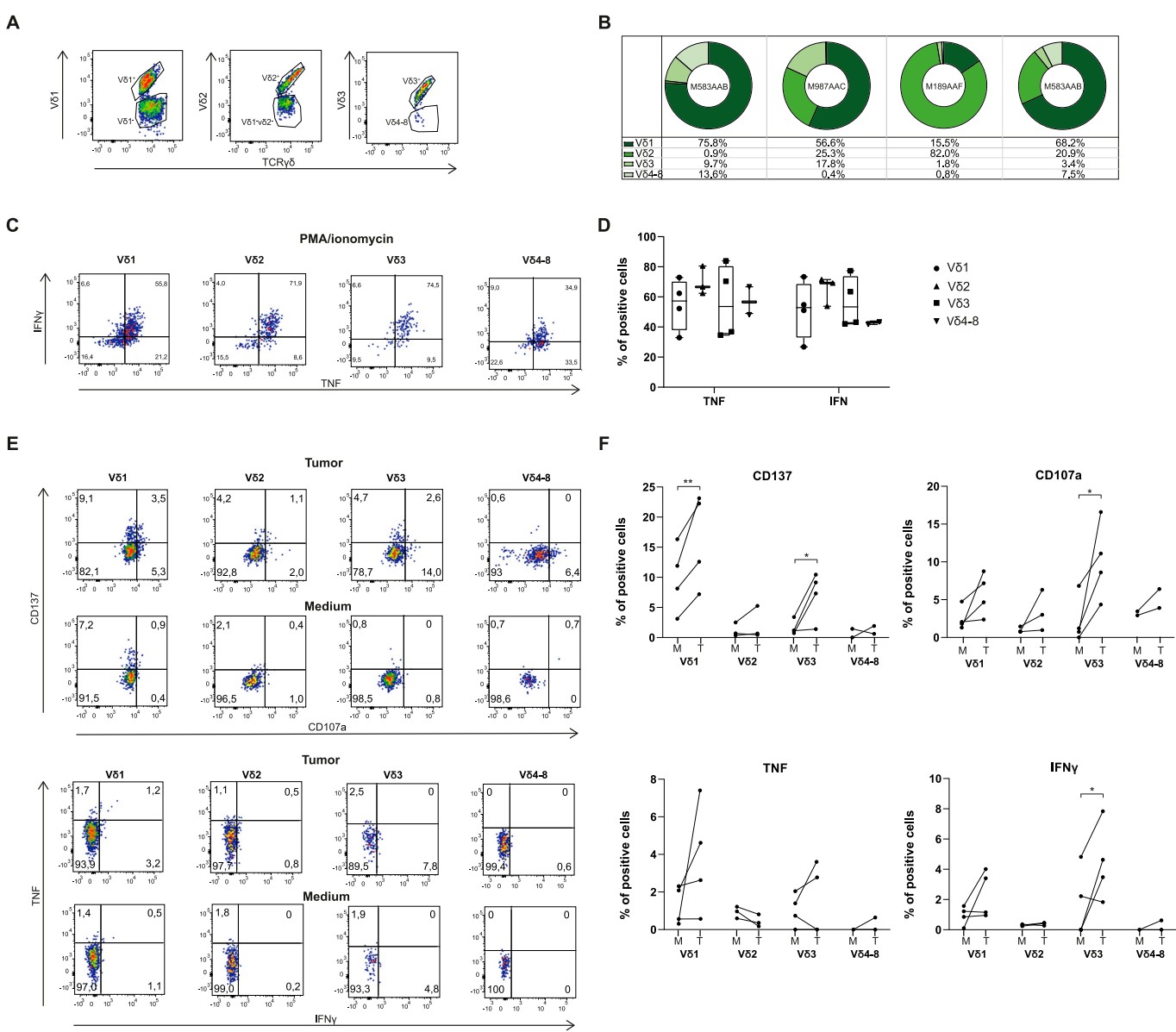

**Figure 4. Tumor-reactive γδ T cells primarily consist of the Vδ1 and Vδ3 subset.**
**(A)** Representative dot plot for Vδ1, Vδ2, Vδ3, and Vδ4-8 tumor-infiltrating lymphocytes (TILs), pregated on γδ T TILs (sample M987AAC). **(B)** Distribution of Vδ1, Vδ2, Vδ3, and Vδ4-8 as the percentage of total γδ T-cell population in four expanded TIL products. γδ T-cell subsets were excluded from further analysis when cell numbers were <100 cells. **(C)** Representative FACS plots for cytokine secretion by Vδ1, Vδ2, Vδ3, and Vδ4-8 TILs after stimulation for 6–7 h with PMA/ionomycin. **(D)** Compiled data (n = 4). **(E, F)** Representative FACS plots and (F) compiled data of the indicated marker on Vδ1, Vδ2, Vδ3, and Vδ4-8 TILs after 6–7 h of coculture with autologous tumor digest in a 1:1 ratio. Each dot represents one TIL product. Box-and-whisker plots depict median, minimum, and maximum values (whiskers) and 25th and 75th pct (box). Wilcoxon's test; * < 0.05, ** < 0.01.

expansion of γδ T cells, we here show that for pediatric γδ T cells, IL-2 suffices for expansion. Neuroblastoma patients are overall young and mostly below the age of 10, and this may therefore support the effective expansion of γδ T cells with IL-2 alone in our study. In sum, we provide first insights into the difference of adult versus pediatric solid tumor-derived γδ T cells. Based on our proof-of-concept study, we propose that γδ T cells should be considered for treating pediatric solid tumors, as we showcased here for the feasibility to develop autologous, tumor-reactive TIL products.

# Materials and Methods

## Patient characteristics

Between February 2020 and January 2022, 19 neuroblastoma patients aged 8–139 mo (average: 49.6 mo), 11/19 (58%) female, were included in this study. One patient sample (M502AAD) was excluded because of the suspicion of not containing exclusively tumor material but being a lymph node and therefore containing

many non–tumor-specific immune cells (Fig S1C). The included patients' characteristics, origin of tumor, and pretreatment regimens are reported in Table S1. Patients were categorized as high risk (HR; n = 12), medium risk (MR; n = 5), and low risk (LR; n = 1). Tumors were resected from adrenal gland (n = 11), soft tissue (n = 4), or other organs (n = 3). One patient (M635AAA) was included in the observation group and received no treatment before debulking surgery. All other patients had received additional treatment before resection, including standard chemotherapy (n = 11), combinations of different chemotherapy regimens (n = 5), combination of chemotherapy with MIBG (n = 2), or chemotherapy followed by autologous stem cell transplantation (n = 1). The study was conducted according to the guidelines of the Declaration of Helsinki and approved by the Institutional Review Board (or Ethics Committee) of the Princess Maxima Centrum for Pediatric Oncology (Utrecht, The Netherlands) (protocol code MEC-2016-739, and date of approval 13 December 2016). Informed consent was obtained from all subjects involved in the study.

The adult tumor samples depicted in Fig S2C were obtained according to the Declaration of Helsinki (seventh revision, 2013), with consent of the Institutional Review Board of the Netherlands Cancer Institute-Antoni van Leeuwenhoek Hospital (NKI-AvL), Amsterdam, The Netherlands (study number CFMPB317). All subjects provided written informed consent.

### Sample collection

Tumor tissue was obtained directly after surgery and processed within 4 h. Tumor samples (n = 20; from patient M909AAA and M189AAF, two samples from different tumor regions were included) were minced into pieces of 1 $mm^3$ and digested with collagenase IV (Worthington) for max 1 h at 37°C and filtered over a 70-$\mu$m cell strainer to obtain a single-cell suspension. Tumor digest was frozen in liquid nitrogen in 90% FCS/10% DMSO (Corning) until further use.

### TIL expansion

Tumor samples were thawed in 50 ml warm RPMI medium (Gibco) supplemented with 2% FCS (Bodego, Bodinco BV), washed, and cultured in 20/80 T-cell mixed media (Miltenyi) containing 5% FCS, 5% human serum (HS) (Sanquin), 1.25 mg/ml fungizone, and 6,000 IU/ml IL-2 (Proleukin, Novartis). Depending on cell density and cloudiness, cells were cultured in a 24-well (1 ml), 48-well (500 $\mu$l), or 96-well (200 $\mu$l) plate at 37°C and 5% $CO_2$ for 12–14 d, with approximately at a concentration of one million cells per ml. Medium was usually refreshed on days 6, 9, and 12, and cells were split when a monolayer of cells was visible in the entire well. After 12–14 d, cells were expanded using a down-scaled minor adjusted version of the clinically approved REP (Rohaan et al, 2022). Briefly, cells were collected and manually counted (hemocytometer) with trypan blue solution (Sigma-Aldrich). 300,000 viable cells (100,000/well) were cocultured with $5 \times 10^6$–$10 \times 10^6$ irradiated PBMCs pooled from 15 healthy donors (feeder cells) in a 24-well plate, containing 30 ng/ml anti-CD3 antibody (OKT-3; Miltenyi Biotec) and 3,000 IU/ml hrIL-2 for 12–14 d at 37°C and 5% $CO_2$. Medium was refreshed every other day, and cells were split when a monolayer in the entire

well was achieved. After 12–14 d, cells were harvested and manually counted. Expanded cells were cryopreserved in RPMI-1640 medium containing 10% DMSO, 40% FCS until further use.

### Tumor reactivity assay

From the well-expanded TIL products, cryopreserved tumor digest and REP TILs were thawed and incubated in 20/80 T-cell mixed media containing 5% FCS, 5% HS, 1.25 mg/ml fungizone, and 500 IU/ml IL-2 overnight at 37°C to recover from thawing. In addition, tumor digest was stimulated with 100–1,000 U/ml IFNγ (Bio-Techne, R&D Systems) overnight, to increase HLA expression. The next day, cells were counted manually with trypan blue solution. A total of $1 \times 10^5$ live expanded TILs were cocultured with $1$–$2 \times 10^5$ live tumor digest cells for 7 h at 37°C. As controls, $1 \times 10^5$ live expanded TILs were stimulated with 10 ng/ml PMA (Sigma-Aldrich) and 1 $\mu$g/ml ionomycin (Sigma-Aldrich) or were cultured with T-cell mixed media only. After 1 h of coculture, 1x brefeldin A (Invitrogen) and 1x monensin (Invitrogen) and anti-CD107a BUV395 (BD Biosciences) were added.

From three patient tumor digest samples, adherent cell cultures could be successfully grown; however, these tissue cell cultures did not contain tumor cells based on single nucleotide polymorphism (SNP) data. Expanded TILs were cocultured with single-cell non-tumorous tissue cells pretreated overnight with hrIFNγ.

### Flow cytometry analysis

Flow cytometry analysis was performed on defrosted and washed material. Tumor digest ex vivo was stained with Fixable Viability Dye eFluor 506 (65-0866-14; Thermo Fisher Scientific) for 20 min at 4°C. After washing, cells were stained with the fluorescently labeled surface antibodies (Table S2) in the dark for 20 min at 4°C. After surface staining, cells were fixed and permeabilized using the FOXP3/transcription factor staining buffer set (Invitrogen) and subsequently stained with intracellular antibodies for 30 min at 4°C in the dark. Defrosted PBMCs from 13 adult healthy donors were used for staining controls. Samples were acquired on an Aurora flow cytometer (Cytek Biosciences) with SpectroFlo software.

Flow cytometry analysis of expanded TIL products was performed with defrosted material. Cells were stained with antibodies against CD3, CD4, CD8, CD56, TCRγδPAN, and in addition with Vδ1, Vδ2, and Vδ3 for analysis of γδ T-cell subsets in a separate experiment. For phenotypic analysis, antibodies against CD11c, CD16, CD19, CD20, CD25, and CD45 were added as well for 30 min at 4°C in the dark (Table S2). Near IR was added for dead cell exclusion. Cells were washed and fixed with the Perm/Fix Foxp3 staining kit (Invitrogen) according to the manufacturer's protocol. Cells were stained with antibodies against TNF, IFN, CD137, and IL-2 (T-cell activation) or Foxp3 (phenotypic analysis) for 30 min at 4°C in the dark. Cells were washed twice and passed through a 70-$\mu$M single-cell filter before acquisition with the Symphony A5 flow cytometer (BD Biosciences) or the ID7000 spectral cell analyzer (Sony Biotechnologies). A standardized cryopreserved PBMC sample pooled from four healthy donors was included as a control for each measurement. Flow cytometry settings were defined for each

patient with single antibody staining. Cell subsets were excluded for further analysis when the cell numbers were below 20 cells (ex vivo) or 100 cells (post-REP); the population was assigned as 0%. Data analysis was carried out with FlowJo Star 10.7.1 (BD).

### Immunohistochemistry

Immunohistochemistry of human neuroblastoma paraffin-embedded tissue slides with hematoxylin and eosin and anti-CD3 was performed by the Pathology Department of the Princess Máxima Center, Utrecht. Tumor tissue slides were pretreated with a citrate solution for 20 min at 100°C, and incubated with the CD3 antibodies for 15 min at RT. The staining was performed on a BOND immunostainer and visualized with the BOND polymer refine detection kit with a DAB enhancer. Analysis of the IHC stainings was performed using QuPath, using automated cell detection with adjusted parameters and subsequent automated cell classification based on CD3 intensity. The number of CD3$^+$ cells per mm$^2$ was used as a metric to quantify the amount of CD3$^+$ cells infiltrating the tumor lesions.

### Statistical analysis

Statistical analysis was carried out with GraphPad Prism 8.0.2 (Dotmatics). Data are shown as single data points with box and whiskers showing maximum, 75th percentile, median, 25th percentile, and minimum, or as paired data points for each patient sample. Significance was calculated with a nonparametric paired *t* test with a two-tailed *P*-value (Wilcoxon's test), or with Tukey's multiple comparison test; variance was calculated as SD. The *P*-value cutoffs were set as * < 0.05, ** < 0.01, *** < 0.001, **** < 0.0001. If not significant, the *P*-value is not shown. Correlation plots were corrected for multiple comparison (n = 6) using the Bonferroni method.

## Data Availability

Because of privacy regulations, data are available upon reasonable request. Please contact MC Wolkers (m.wolkers@sanquin.nl).

## Supplementary Information

## Acknowledgements

We like to thank the Flow Cytometry Facility staff and the Department of Cryobiology from Sanquin, and the nursing staff and other employees involved from the Princess Máxima Center. We thank Dr. John Haanen, Dr. Koen Hartemink, Dr. Kim Monkhorst, and Dr. Axel Bex from the Antonius van Leeuwenhoek ziekenhuis-Netherlands Cancer Institute for supporting our studies in adult solid tumors, and Aurélie Guislain for technical support. We also thank all patients and their parents for their willingness to contribute to science. This study was supported by an internal grant of Sanquin (PPOC 21-07) to MC Wolkers, by Oncode Institute to MC Wolkers, and by Stichting Kinder Kankervrij (KIKA 404 and KIKA 491) to J Wienke and MC Wolkers. The authors have no conflicting financial interests.

### Author Contributions

SM Castenmiller: conceptualization, formal analysis, and writing—original draft, review, and editing.
AL Borst: data curation, formal analysis, and writing—review and editing.
L Wardak: data curation and writing—review and editing.
JJ Molenaar: conceptualization.
M Papadopoulou: methodology and writing—review and editing.
RR de Krijger: project administration and writing—review and editing.
AFW van der Steeg: project administration and writing—review and editing.
MM van Noesel: project administration and writing—review and editing.
D Vermijlen: methodology and writing—review and editing.
R de Groot: conceptualization and writing—review and editing.
J Wienke: conceptualization, data curation, formal analysis, and writing—review and editing.
MC Wolkers: conceptualization, supervision, and writing—original draft.

### Conflict of Interest Statement

The authors declare that they have no conflict of interest.

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
