## [Reviewer comments · Life Science Alliance]

Life Science Alliance

$\gamma\delta$ T cells are the prime anti-tumoral T cells in pediatric neuroblastoma

Suzanne Castenmiller, Anne Borst, Leyma Wardak, Jan Molenaar, Maria Papadopoulou, Ronald de Krijger, Alida van der Steeg, Max van Noesel, David Vermijlen, Rosa Groot, Judith Wienke, and Monika Wolkers

DOI: <https://doi.org/10.26508/lsa.202503249>

Corresponding author(s): *Monika Wolkers, Sanquin*

Review Timeline:

Submission Date:	2025-02-05
Editorial Decision:	2025-03-31
Revision Received:	2025-07-10
Editorial Decision:	2025-08-13
Revision Received:	2025-08-20
Accepted:	2025-08-21

Scientific Editor: Sarita Hebbar

Transaction Report:

March 31, 2025

Re: Life Science Alliance manuscript #LSA-2025-03249-T

Dr. Monika C Wolkers
Sanquin Research Institute
Hematopoiesis
Plesmanlaan 125
Plesmanlaan 125
Amsterdam 1066CX
Netherlands

Dear Dr. Wolkers,

Thank you for submitting your manuscript entitled " $\gamma\delta$ T cells are the prime anti-tumoral T cells in pediatric neuroblastoma" to Life Science Alliance. The manuscript was assessed by three expert reviewers, whose detailed comments are appended to this letter.

Overall, the reviewers found this work of potential value to the community but have highlighted the need for major revisions to substantiate the conclusions in the manuscript. The areas of concern include (1) detailed characterisation the primary tissue, (2) identification of tumour-reactive $\gamma\delta$ T cells before expansion, and (3) variability of T-cell expansion. Reviewers 2 and 3 raised an important point for the identification of biases at each step of the workflow, and to strengthen the manuscript with an extended characterisation and/or details. Finally all the reviewers suggested points for elaboration in the discussion.

Given the overall recommendations, we would like to invite you to submit a revised manuscript addressing the Reviewers' comments. Please include a letter addressing all the reviewers' comments point by point.

Thank you for this interesting contribution to Life Science Alliance. We are looking forward to receiving your revised manuscript.

Sincerely,

Sarita Hebbar, PhD
Scientific Editor
Life Science Alliance
<http://www.lsajournal.org>

B. MANUSCRIPT ORGANIZATION AND FORMATTING:

Reviewer #1 (Comments to the Authors (Required)):

This study provides evidence that $\gamma\delta$ T cells are the primary anti-tumoral immune cells in pediatric neuroblastoma. These cells can be efficiently expanded and exhibit strong tumor-reactive properties, opening new avenues for immunotherapy. The unique immune environment of pediatric neuroblastoma may enable $\gamma\delta$ T cells to serve as a promising target for future therapeutic strategies.

Major comments:

1. While $\gamma\delta$ T cells expanded well in neuroblastoma TILs, it is unclear why this expansion is successful in pediatric tumors but not in adult tumors. Can the authors explore age-related differences? Could additional cytokines or co-stimulatory factors enhance $\gamma\delta$ T cell expansion? Can age-related differences be studied via in vitro settings (e.g. anti-tumor reactivity against neuroblastoma cell lines of pediatric vs adult lymphocytes)?
2. The study tested PD-1 and CD137 as potential markers of tumor-reactive T cells, but these did not correlate well with actual tumor-reactivity. This raises questions about how to identify tumor-reactive $\gamma\delta$ T cells before expansion, and it would be good to discuss these issues.
3. It would be interesting to also investigate the impact of the tumor microenvironment on $\gamma\delta$ T cell expansion, presumably via in vitro experiments.
4. The TIL expansion were very variable. Could the authors explain this better by the mechanism of tumor recognition? Also, a standardized quality control for starting material could help clarify if the variability is due to tumor-intrinsic factors or methodology. Moreover, the TIL expansion was done after cryopreservation which can lead to cell loss, and hence my explain the variability. Could the authors control this?
5. Overall the study is very descriptive. In vitro experiments of how the expanded TILs function in neuroblastoma would enhance the impact of the study.

Reviewer #2 (Comments to the Authors (Required)):

In their study, Castenmiller et al. studied the expansion of T cells from neuroblastoma (NB) tissue samples. The authors were able to show that tumor digest contained sufficient T cell for expansion, and that the Rapid Expansion Protocol was effective. Of interest, the frequency of $\gamma\delta$ T cells remained unchanged, which is usually not seen with the expansion of TILs from tumors of adult patients. Remarkably, $\gamma\delta$ T cell showed the strongest activation by incubation with tumor digest suggesting that they are very important for anti-tumoral T cell responses in neuroblastoma.

Immunotherapy of neuroblastoma remains a challenge. Thus, the study provides important insights for the field given that working with primary neuroblastoma tissue is a challenge by its own in terms of handling and access to the material. Several issues remain to be addressed.

1. The authors report a relatively high frequency of immune cells in their tumor digests (Fig.1C). From this they conclude that NB isn't as 'cold' as commonly believed. However, the flow cytometry may be subject to biases such as differential viability/fragility/vulnerability of NB cells and other cell types, whereas immune cells are well preserved for flow cytometry. Actually, this is quite common with tumor digests (from other cancer types) - at least based on own experiences. Though this is not a critical issue for the conclusion of the paper, it would strengthen the manuscript to compare it to matched IHCs for CD45, CD3. Also, a more critical discussion should be added to the text.

2. In Fig. 3B, gd T cells show quite a significant level of activation in the medium condition. How can this be explained? Does it impact on the central conclusion that gd T cells are the prime anti-tumoral T cells in NB.

3. A flow-chart of the study outline would be helpful including patient characteristics. How many patients were included, which cases dropped out before the final analyses in Fig. 3 and for what reasons.

4. Paired t-test statistics are used for the comparison of the frequencies, but there is the issue of 0 and 1 boundaries and the reported values are closed to 0. Transformations like arcsine square root transformation or the use of non-parametric test would be more appropriate.

Reviewer #3 (Comments to the Authors (Required)):

Castenmiller et al. describe a potential novel therapy approach for neuroblastoma, an aggressive pediatric solid tumor, using ex vivo expanded tumor infiltrating lymphocytes (TILs). Evidence is presented that TILs are present in primary neuroblastoma tumors, even following exposure to standard treatment approaches including chemotherapy and radiation (MIBG). The TILs present were expanded ex vivo using IL-2 as stimuli under the REP protocol and produced expanded TIL products containing CD8+ T, conventional (CD4+) T, and $\gamma\delta$ T cells. Unlike adult solid tumors, $\gamma\delta$ T cells expanded with similar efficacy to $\alpha\beta$ T cells. Evidence is provided that $\gamma\delta$ T cells may drive autologous anti-tumor activity.

The authors present a compelling story for further investigation of TIL-based adoptive cell therapy approaches for neuroblastoma (and potentially other pediatric solid tumors), with specific focus on the importance and anti-neuroblastoma activity of $\gamma\delta$ T cells. It is important to highlight that the authors state that this study provides the first evidence that TIL products can be generated from pediatric neuroblastoma tumor lesions. However, this statement appears inaccurate as it overlooks at least one prior report of TIL products expanded from primary neuroblastoma tumors already in the literature. While previous publications may decrease the novelty of the work from that aspect, there are other intriguing aspects of the manuscript that describe new findings. Of particular interest, the investigations performed highlight differences between pediatric and adult TIL products, most specifically differences in the $\gamma\delta$ T cell compartment that are potentially related to differences in $\gamma\delta$ T cell biology due to age. While differences between pediatric and adult $\gamma\delta$ T cells have recently been described under healthy conditions, this would be one of the first investigations in this topic in the setting of neuroblastoma.

The manuscript appears to be well-suited for the journal, Life Science Alliance, both in subject material and impact. While the article is generally well-written, the story is underdeveloped and would benefit from major revisions, including additional experimentation to more thoroughly develop and confirm the manuscript's most prominent findings.

Points of Major Revision/Clarification

Figure 1:

1. Tumors were digested with collagenase IV for up to 1 hour to achieve single cell suspensions. While likely a gentler method than other commonly used approaches, collagenase IV treatment has been shown to disrupt epitopes critical for detection of cell surface markers, including CD56, and may decrease sample viability at exposures >30 minutes (Polakova et al 2023). Did the authors confirm that collagenase treatment did not impact markers used within their panel for immunophenotyping? One consideration would be to show the panel staining comparison on healthy donor control PBMCs +/- collagenase as a control that staining is at least similar. Alternatively, in the reviewer's experience simple non-enzymatic, mechanical dissociation using the miltenyi C-tube platform produces digested samples from primary neuroblastoma tumors with similar viability to this study. Could the authors digest and stain a tumor in this fashion and show comparable staining. Either of these methodology controls would help to identify limitations in detections and be important for interpretation of results - for example, were NK/NKT numbers truly low or were CD56 and CD16 staining greatly impacted by the processing protocol?

2. The cell numbers for the representative tumor in 1A appear to be sufficient to produce very robust flow plots for downstream characterization. Was this the case for all tumors? It would be ideal to include event numbers within each gate for each sample to enhance the reliability of the findings (or at least a summary) and/or define the minimum number of viable events produced by

tumors (and if recorded, how this corresponded with viable cell number post processing). In methods, the authors state that cell subsets were excluded for further analysis if cell numbers were below 100 cells - could they provide an estimate of how often this occurred? For plots D and E, it appears this occurrence were rare as most plots show 20 dots or did the authors assign those values as 0% and plot in that fashion? Additional clarity here would be appreciated to ensure robustness of results. Did the authors investigate the abundance of different $\gamma\delta$ T subclasses (V δ 1, V δ 2, V δ 3, etc?) within all 20 tumor specimens (not clear from methods - "with V δ 1, V δ 2, and V δ 3 for analysis of $\gamma\delta$ T subsets in a separate experiment")? If so, please include this characterization of the primary tumor tissue prior to expansion.

3. It is appreciated that the authors attempted to correlate tumor immune composition with disease stage and treatment regimens. A step further, did the authors investigate how immune composition of the primary (debulked) tumor related therapy response? While it is a mixed cohort regarding disease stage and therapy received, this is a decently large patient cohort, and it would be interesting to investigate if higher percentages of specific cell subsets corresponded with response to therapy and/or short-term survival. Tissue collection ended January 2022 so some survival data should/might? be available.

Figure 2/Supplemental Figure 1:

1. In the representative gating for REP-expanded TILs, it appears that Live/Dead gating was performed based on FSC vs. SSC - was this indeed the case or was a live dead stain included. If not, why was the methodology for back gating changed considering Figure 1 includes Zombie viability staining?

2. Can the authors provide additional details for the ex vivo and postREP adult tumors depicted in 2H? These are not described in methodology. How many tumors were included? Were the adult tumors also primarily post-treatment? Was expansion and overall viability similar to neuroblastoma?

Supplemental Figure 2:

1. Please provide IFN γ /TNF/IL-2 staining for unstimulated CD8/Tconv/ $\gamma\delta$ T for comparison to PMA/ionomycin stimulated cells in Supplemental Figure 2A.

2. In lines 196-197, it seems to be downplayed that PD-1 is significantly upregulated on CD4+ T cells of TILs ex vivo. Additionally, in Figure 3 TNF α is significantly upregulated in CD4+ T cells following co-culture. The potential activity and role of CD4+ T cells should therefore also be discussed more.

Figure 3:

1. Did the authors consider any other markers for T cell activation (i.e., CD69 or CD25)? These may be more broadly accepted as classical markers of activation compared to CD137.

2. The representative results for CD137 expression do not seem to as robustly capture CD137 upregulation as is described in text and quantified in Figure 3B. Is this the best representative image?

3. Can the authors define why $\gamma\delta$ T cells would show activation against non-tumorous (NT) tissue in 3C? If it is expected that only tumor-reactive TILs were expanded via the REP protocol, why would there be activation against non-tumor tissue? Could this be a potential therapeutic safety issue? Please address if not already covered in the discussion.

Figure 4:

1. Please provide IFN γ /TNF staining for unstimulated $\gamma\delta$ T subsents for comparison to PMA/ionomycin stimulated cells in Figure 4C/D.

2. Were statistics performed for comparisons provided in 4F? Without any statistical comparisons, it is hard to confirm the findings reported in lines 214-220 and the conclusion that V δ 1 and V δ 3 are the prime $\gamma\delta$ T cell subsets responding to autologous tumor digests. This should also be addressed in the abstract, which without any statistical confirmation, overstates the role of V δ 1 and V δ 3 Ts (Line 36).

Discussion:

1. Lines 222-223 state, "In summary, we here provide first evidence that TIL products can be generated from pediatric neuroblastoma tumor lesions." This statement may not be entirely accurate considering at least one previous literature report that demonstrates TIL expansion from primary neuroblastoma tissues (Hurtado...Anderson et al, 2019). Please adjust accordingly and also include description of how findings of this report substantiate or differ from the previous work. Interestingly, this report also provides description of intratumoral $\gamma\delta$ Ts.

2. One very interesting aspect of the manuscript is the differences between pediatric and adult $\gamma\delta$ T cells and their presence/expansion/activity against neuroblastoma tumors both as TILs and potentially as ACT approaches. It would be nice if the authors could perform additional experiments to further develop this aspect of their story.

Points of Minor Revision:

1. In Figure 1, depiction of the gating strategy could be clarified for audiences with limited phenotyping experience. For example, the authors should include clarity that the NK cell gate (A, bottom row, farthest left) originated from the parental CD3- gate (A, top row, farthest right) and that remaining T cell gates (A, bottom row, plots 2-3) originated from parental CD3+ $\gamma\delta$ T- gate (A, top row, farthest right). If the assumed gating strategy, described above, was not indeed the gating tree the authors should provide further clarification within the figure.

2. While the text is clear regarding the expansion components and timing of pre-REP/REP phases, some readers might appreciate a diagram depicting the expansion protocol used within Figure 2 as visual summary. Additionally, it could be helpful to include a legend of what black (expansion >500-fold) vs. red (expansion <500-fold) dots symbolize within the figure itself.

3. It is a somewhat confusing that the summary table provided in Supplemental Figure 1 provides mean but the corresponding plots in Figure 2 use median. In the text average is provided - providing some sort of uniformity between the three could make the results easier to follow.
4. Panels and text for Figure 2A,C-F and Supplemental Figure 1D are somewhat hard to follow. Data is repeated (for purpose of different comparisons) but this isn't immediately clear. Consider revising for clarity or condensing into one figure if possible?

Reviewer #1 (Comments to the Authors (Required)):

This study provides evidence that $\gamma\delta$ T cells are the primary anti-tumoral immune cells in pediatric neuroblastoma. These cells can be efficiently expanded and exhibit strong tumor-reactive properties, opening new avenues for immunotherapy. The unique immune environment of pediatric neuroblastoma may enable $\gamma\delta$ T cells to serve as a promising target for future therapeutic strategies.

Major comments:

1. While $\gamma\delta$ T cells expanded well in neuroblastoma TILs, it is unclear why this expansion is successful in pediatric tumors but not in adult tumors. Can the authors explore age-related differences? Could additional cytokines or co-stimulatory factors enhance $\gamma\delta$ T cell expansion? Can age-related differences be studied via in vitro settings (e.g. anti-tumor reactivity against neuroblastoma cell lines of pediatric vs adult lymphocytes)?

We agree with the reviewer that the different outcomes of $\gamma\delta$ TIL expansion between pediatric and adult tumors is intriguing. scRNA seq data from neuroblastoma $\gamma\delta$ T cells reveal that in children, $\gamma\delta$ T cells express similar levels of the IL-2 receptor alpha and beta mRNA as $\alpha\beta$ T cells (**NEW Figure S2A**), suggesting that the IL-2 receptor expression renders pediatric $\gamma\delta$ T cells responsive to the IL-2 included in the TIL culture medium. Adult $\gamma\delta$ T cells were shown to respond less to IL-2 stimulation alone when compared to $\alpha\beta$ T cells. Rather, they need additional stimuli for successful $\gamma\delta$ T cell expansion (Verkerk et al. Front Immunol, 2024. Almeida et al. Clin Cancer Res, 2016).

To date, we do not know at which age the capacity to expand is lost for $\gamma\delta$ T cells, as suggested by literature (Clark et al. Int J Mol Sci, 2020. De Rosa et al, J Immunol, 2004. Schatorjé et al, Scand J Immunol, 2012). In our patient cohort, we did not find correlations of expansion potential with the increase of age (see Figure 2B), suggesting that $\gamma\delta$ T cells lose the responsiveness to IL-2 later than 10 years of age. We have now included these considerations in the discussion (**page 9**).

As to testing additional cytokines or co-stimulatory factors to further enhance $\gamma\delta$ T cell expansion, this is indeed our current focus. In this patient cohort, tumor material was limited. We could therefore not yet perform the appropriate comparisons (of note: due to high inter-well heterogeneity in TIL cultures, at least 4 wells/condition are required for comparative studies so that inter-well variations can be disentangled from actual cytokine/co-stimulatory signals). In pilot experiments, we learned that the optimizations tested with adult blood-derived T cells did not translate to pediatric tumor-derived $\gamma\delta$ T cells. In conclusion, while we are currently working on improving TIL cultures, it is not as straight-forward as we hoped, and we therefore consider these efforts outside the scope of this proof-of-concept manuscript.

NEW Figure S2A.

2. The study tested PD-1 and CD137 as potential markers of tumor-reactive T cells, but these did not correlate well with actual tumor-reactivity. This raises questions about how to identify tumor-reactive $\gamma\delta$ T cells before expansion, and it would be good to discuss these issues.

Indeed, PD-1 and CD137 do not correlate well with the tumor-reactivity of the $\gamma\delta$ T cells, and we agree with the reviewer that this is an interesting discussion point. Measuring PD-1 and CD137 was selected based on our knowledge to identify tumor-reactive $\alpha\beta$ T cells. In fact, when we started this study, we were not particularly focussing on $\gamma\delta$ T cells, but rather expected $\alpha\beta$ CD4/CD8 T cells to be tumor reactive, if at all. Therefore, the ex vivo panel was not optimized for defining tumor-reactive $\gamma\delta$ T cells. In future studies, we aim to include more specific $\gamma\delta$ T cell markers (e.g. DNAM and KLRD1) and other checkpoint receptors. We highlighted this point in the discussion on **page 9**.

3. It would be interesting to also investigate in the impact of the tumor microenvironment on gd T cell expansion, presumably via in vitro experiments.

We agree with the reviewer. However, as tumor material from the children is very limited, we unfortunately cannot provide such data at this stage. We have discussed it however on **page 9**.

4. The TIL expansion were very variable. Could the authors explain this better by the mechanism of tumor recognition? Also, a standardized quality control for starting material could help clarify if the variability is due to tumor-intrinsic factors or methodology. Moreover, the TIL expansion was done after cryopreservation which can lead to cell loss, and hence my explain the variability. Could the authors control this?

Due to the scarce availability and limited starting material, a standardized quality control for the starting material was not possible. To date, the best prediction for good TIL expansion is the amount of starting material, as we report in **Figure 2B, right panel**. We have highlighted this point better in the discussion (**page 10**).

As to cryopreservation: we fully agree that fresh material would be preferable. For this proof-of-concept study, it was unfortunately not feasible. The prevalence of high-risk neuroblastoma in the Netherlands is low, and collecting fresh material would have taken years. Using frozen material allowed us to include historical samples and thus substantially speeded up the data collection. However, as the reviewer correctly points out, we have learned ourselves from the adult TIL cultures that fresh material (starting cultures straight after surgery) is more efficient. Provided that we already obtained promising results with frozen material, we postulate that the quality of TIL products should be even better once the logistics are set up for this translational approach with fresh material. We have now highlighted this important point in the discussion (**page 9**).

5. Overall the study is very descriptive. In vitro experiments of how the expanded TILs function in neuroblastoma would enhance the impact of the study.

Please be advised that this study is a proof-of-concept study, which was set up to test whether TIL therapy is a viable option for neuroblastoma patients. We agree that many other assays are needed to improve $\gamma\delta$ TIL expansion and to assess their functionality, as we already indicated in the discussion (**page 9**). To date, measuring cytokine production profiles is the current state-of-the-art analysis of tumor-reactive TILs being present in expanded TIL products.

Reviewer #2 (Comments to the Authors (Required)):

In their study, Castenmiller et al. studied the expansion of T cells from neuroblastoma (NB) tissue samples. The authors were able to show that tumor digest contained sufficient T cell for expansion, and that the Rapid Expansion Protocol was effective. Of interest, the frequency of gd T cells remained unchanged, which is usually not seen with the expansion of TILs from tumors of adult patients. Remarkably, gd T cell showed the strongest activation by incubation with tumor digest suggesting that they are very important for anti-tumoral T cell responses in neuroblastoma. Immunotherapy of neuroblastoma remains a challenge. Thus, the study provides important insights for the field given that working with primary neuroblastoma tissue is a challenge by its own in terms of handling and access to the material. Several issues remain to be addressed

We are very pleased that the reviewer appreciates the novelty and importance of our findings for developing immunotherapy for neuroblastoma. Please find our response to the questions below.

1. The authors report a relatively high frequency of immune cells in their tumor digests (Fig.1C). From this they conclude that NB isn't as 'cold' as commonly believed. However, the flow cytometry may be subject to biases such as differential viability/fragility/vulnerability of NB cells and other cell types, whereas immune cells are well preserved for flow cytometry. Actually, this is quite common with tumor digests (from other cancer types) - at least based on own experiences. Though this is not a critical issue for the conclusion of the paper, it would strengthen the manuscript to compare it to matched IHCs for CD45, CD3. Also, a more critical discussion should be added to the text.

We have now included IHC stainings for three representative tumor samples with low (<10%), medium (10-30%) or high (>30%) percentage of CD3⁺ T cell infiltrates, as defined by flow cytometry. Even though the sections used for IHC and for TIL expansion stem from different parts of the same tumor, we found that the number of CD3⁺ T cells counted in IHC slides reflected well the percentage of CD3⁺ T cells defined by flow cytometry (**NEW supplemental figure S1D**).

NEW Figure S1D

2. In Fig. 3B, $\gamma\delta$ T cells show quite a significant level of activation in the medium condition. How can this be explained? Does it impact on the central conclusion that $\gamma\delta$ T cells are the prime anti-tumoral T cells in NB.

Indeed, $\gamma\delta$ T cells display a level of activation in the medium condition already. Albeit not as strong, we observe similar effect on $\alpha\beta$ T cells in TIL products generated from Renal cell carcinoma (van Asten, Oncoimmunology, 2021). To date, we can only speculate about the underlying reason. A part of it may stem from the fact that TILs are cultured for 4 weeks with high dose IL-2, resulting in highly reactive effector T cells.

More importantly, to determine the anti-tumor response of $\gamma\delta$ T cells, we determined the increase of CD137 between the medium condition (e.g. basal level) and autologous tumor digest. CD137 was previously used to detect $\gamma\delta$ T cell responses to infections (Ji, 2024. Pei, 2020), which we now highlight in the discussion on **page 10**. Exposure to tumor cells further increased the expression of CD137 and the other activation markers when compared to medium control. To decipher whether this was merely due to the addition of cells, we tested the fibroblast cells grown from neuroblastoma patients. Again, higher responses were found in co-cultures with tumor cells (Figure 3C). We thus conclude from these findings that $\gamma\delta$ T cells are tumor-reactive.

3. A flow-chart of the study outline would be helpful including patient characteristics. How many patients were included, which cases dropped out before the final analyses in Fig. 3 and for what reasons.

This is a good suggestion and we included a scheme depicted below as **NEW Figure S1C**.

4. Paired *t*-test statistics are used for the comparison of the frequencies, but there is the issue of 0 and 1 boundaries and the reported values are closed to 0. Transformations like arcsine square root transformation or the use of non-parametric test would be more appropriate.

We agree, and we adjusted the analyses to non-parametric paired t tests (Wilcoxon test) throughout. Importantly, even though the numbers slightly change (new values indicated in red in the revised manuscript), all conclusions of the results remain.

Reviewer #3 (Comments to the Authors (Required)):

Castenmiller et al. describe a potential novel therapy approach for neuroblastoma, an aggressive pediatric solid tumor, using ex vivo expanded tumor infiltrating lymphocytes (TILs). Evidence is presented that TILs are present in primary neuroblastoma tumors, even following exposure to standard treatment approaches including chemotherapy and radiation (MIBG). The TILs present were expanded ex vivo using IL-2 as stimuli under the REP protocol and produced expanded TIL products containing CD8+ T, conventional (CD4+) T, and $\gamma\delta$ T cells. Unlike adult solid tumors, $\gamma\delta$ T cells expanded with similar efficacy to $\alpha\beta$ T cells. Evidence is provided that $\gamma\delta$ T cells may drive autologous anti-tumor activity.

The authors present a compelling story for further investigation of TIL-based adoptive cell therapy approaches for neuroblastoma (and potentially other pediatric solid tumors), with specific focus on the importance and anti-neuroblastoma activity of $\gamma\delta$ T cells. It is important to highlight that the authors state that this study provides the first evidence that TIL products can be generated from pediatric neuroblastoma tumor lesions. However, this statement appears inaccurate as it overlooks at least one prior report of TIL products expanded from primary neuroblastoma tumors already in the literature. While previous publications may decrease the novelty of the work from that aspect, there are other intriguing aspects of the manuscript that describe new findings. Of particular interest, the investigations performed highlight differences between pediatric and adult TIL products, most specifically differences in the $\gamma\delta$ T cell compartment that are potentially related to differences in $\gamma\delta$ T cell biology due to age. While differences between pediatric and adult $\gamma\delta$ T cells have recently been described under healthy conditions, this would be one of the first investigations in this topic in the setting of neuroblastoma.

The manuscript appears to be well-suited for the journal, Life Science Alliance, both in subject material and impact. While the article is generally well-written, the story is underdeveloped and would benefit from major revisions, including additional experimentation to more thoroughly develop and confirm the manuscript's most prominent findings.

We thank the reviewer for this positive evaluation of our study. Please find below our answers to the remaining questions of this reviewer below.

Points of Major Revision/Clarification

Figure 1:

1. Tumors were digested with collagenase IV for up to 1 hour to achieve single cell suspensions. While likely a gentler method than other commonly used approaches, collagenase IV treatment has been shown to disrupt epitopes critical for detection of cell surface markers, including CD56, and may decrease sample viability at exposures >30 minutes (Polakova et al 2023). Did the authors confirm that collagenase treatment did not impact markers used within their panel for immunophenotyping? One consideration would be to show the panel staining comparison on healthy donor control PBMCs +/- collagenase as a control that staining is at least similar. Alternatively, in the reviewer's experience simple non-enzymatic, mechanical dissociation using the miltenyi C-tube platform produces digested samples from primary neuroblastoma tumors with similar viability to this study. Could the authors digest and stain a tumor in this fashion and show comparable staining. Either of these methodology controls would help to identify limitations in detections and be important for interpretation of results - for example, were NK/NKT numbers truly low or were CD56 and CD16 staining greatly impacted by the processing protocol?

We have used collagenase IV already for adult tumor digests (de Groot et al, Oncoimmunology, 2019, Castenmiller et al, ESMO-IOTECH, 2022). Possibly due to the relatively short time of tissue

digestion (20-30 min), the detection of CD56 and CD16 expression was good in our previous studies, and this also holds true for neuroblastoma. Both CD56 and CD16 expression were detected (**NEW Figure S2B**), indicating that the staining was effective after collagenase IV treatment. Please be advised that we always co-stained PBMCs from healthy adult donors in parallel with the identical antibody mix, and we observed the expected CD56+ c.q. CD16+ NK cell percentage in the PBMC samples. Thus, our antibody panel is well suited for the detection of CD56 and CD16 positive NK cells.

NEW Figure S2B

Moreover, not only on protein level, but also on gene expression level, we previously observed lower CD56 (NCAM1) and CD16 (FCGR3A) expression on neuroblastoma-infiltrating NK cells than in peripheral blood NK cells from healthy donors. We therefore believe that the low CD56/CD16 expression on NK cells does not stem from the collagenase treatment but rather is part of a dysfunctional profile, which we described in more detail in Wienke et al Cancer Cell 2023.

2. The cell numbers for the representative tumor in 1A appear to be sufficient to produce very robust flow plots for downstream characterization. Was this the case for all tumors? It would be ideal to include event numbers within each gate for each sample to enhance the reliability of the findings (or at least a summary) and/or define the minimum number of viable events produced by tumors (and if recorded, how this corresponded with viable cell number post processing). In methods, the authors state that cell subsets were excluded for further analysis if cell numbers were below 100 cells - could they provide an estimate of how often this occurred? For plots D and E, it appears this occurrence were rare as most plots show 20 dots or did the authors assign those values as 0% and plot in that fashion? Additional clarity here would be appreciated to ensure robustness of results.

We agree with the reviewer that we could have explained this better. For ex-vivo analysis, we used a cut-off of 20 cells per immune cell subset. None of the cell subsets dropped below this cell number. For postREP analysis, we used a cut-off of 100 cells per immune cell subset. If postREP cell numbers dropped below 100 cells, the percentage was assigned as 0%. We have clarified this in the method section on page 10.

Did the authors investigate the abundance of different $\gamma\delta T$ subclasses (V δ 1, V δ 2, V δ 3, etc?) within all 20 tumor specimens (not clear from methods - "with V δ 1, V δ 2, and V δ 3 for analysis of $\gamma\delta T$ subsets in a separate experiment"? If so, please include this characterization of the primary tumor tissue prior to expansion.

Unfortunately, we cannot provide these data. When we started with TIL products for neuroblastoma and phenotyped the tumor digests, we had not anticipated such an important role of $\gamma\delta$ T cells. We therefore only measured $\gamma\delta$ T cell infiltrates with the pan $\gamma\delta$ TCR antibody ex vivo. Only when we came to realize that $\gamma\delta$ T cells were the responders, we went back and tested the $\gamma\delta$ T cell subsets. However, we could only do so on expanded TILs (Fig 4). So unfortunately, we only know which and in which ratio $\gamma\delta$ T cell subsets are present in the tumors. Based on the $\gamma\delta$ T cell analysis of pediatric tissues specimens on pediatric healthy tissues (Farber, Sci Immunology), we suspect that most $\gamma\delta$ T cells will be non-Vd2 subtype and rather display tissue-resident V subtypes, and that Vd2 cells would be from blood contaminants. We plan, however, to include such analysis in future studies and another cohort with hopefully sufficient material. We will state the importance to extend the analysis on $\gamma\delta$ T cell infiltrates in the tumor (ex vivo) in the discussion (page 9).

3. It is appreciated that the authors attempted to correlate tumor immune composition with disease stage and treatment regimens. A step further, did the authors investigate how immune composition of the primary (debunked) tumor related therapy response? While it is a mixed cohort regarding disease stage and therapy received, this is a decently large patient cohort, and it would be interesting to investigate if higher percentages of specific cell subsets corresponded with response to therapy and/or short-term survival. Tissue collection ended January 2022 so some survival data should/might? be available.

When we correlated the therapy response to clinical characteristics for a different study, we learned that due to the heterogeneity in this cohort in terms of age, sex, treatment regimen, disease stage and environmental factors, no robust correlations could be made. Because the scope of this present study is different, and the question the reviewer poses here is part of a different study, we decided to omit this information in the present study. We hope the reviewer understands this decision.

Figure 2/Supplemental Figure 1:

1. In the representative gating for REP-expanded TILs, it appears that Live/Dead gating was performed based on FSC vs. SSC - was this indeed the case or was a live dead stain included. If not, why was the methodology for back gating changed considering Figure 1 includes Zombie viability staining?

We thank the reviewer for this observation. This is indeed an error, the live/dead gating for REP-expanded TILs was performed with Near-IR dye, which is similar to Zombie dye. This is now corrected in Figure S1C.

2. Can the authors provide additional details for the ex vivo and postREP adult tumors depicted in 2H? These are not described in methodology. How many tumors were included? Were the adult tumors also primarily post-treatment? Was expansion and overall viability similar to neuroblastoma?

We now included a table with clinical characteristics/expansion data for the adult tumor samples in NEW Figure S2C.

Sample	Pre-treatment	Expansion > 2000-fold	Viability >80%
NSCLC1	Naïve	Yes (de Groot, 2020 ¹⁸ , Castenmiller 2021 ¹⁹)	Yes
NSCLC2	Naïve		
NSCLC3	ALK inhibitor		
RCC1	TKI	Yes (van Asten, 2020 ²⁰)	Yes
RCC2	TKI		
RCC3	Naïve		
Melanoma1	Naïve		

Melanoma2	Naïve	Yes (unpublished)	Yes
-----------	-------	-------------------	-----

Supplemental Figure 2:

1. Please provide IFN γ /TNF/IL-2 staining for unstimulated CD8/Tconv/ $\gamma\delta$ T for comparison to PMA/ionomycin stimulated cells in Supplemental Figure 2A.

Please be advised that the unstimulated controls are part of figure 3 A. We now clearly indicated this in the legend of suppl. figure 2, and in the results section with the figure call out (page 6).

2. In lines 196-197, it seems to be downplayed that PD-1 is significantly upregulated on CD4+ T cells of TILs ex vivo. Additionally, in Figure 3 TNFa is significantly upregulated in CD4+ T cells following co-culture. The potential activity and role of CD4+ T cells should therefore also be discussed more.

We fully agree with this reviewer, and had not realized that we had cut too much on our discussion on this indeed important finding during manuscript editing. We now highlighted this finding better in the results section and expanded our discussion on this finding (page 7).

Figure 3:

1. Did the authors consider any other markers for T cell activation (i.e., CD69 or CD25)? These may be more broadly accepted as classical markers of activation compared to CD137.

As indicated above, we agree that it would be highly interesting to further characterize the T cells. Unfortunately, CD25 expression is low ex vivo in all T cell subsets (except from Tregs) and CD69 is high on all, which we rather consider to be a marker for tissue residency in the context of solid tumors. After expansion, CD25 is high on all T cell subsets due to the high IL2 dose during T cell expansion, as is CD69. Due to the 4 week culture with high dose IL-2, we are hesitant to draw any conclusions related to the activation profile of the cells based on CD69 and CD25.

2. The representative results for CD137 expression do not seem to as robustly capture CD137 upregulation as is described in text and quantified in Figure 3B. Is this the best representative image?

We thank the reviewer for this observation. We have adjusted Figure 3A and have chosen a more representative patient sample to show CD137 upregulation. We have adjusted the figure legend of Figure 3 on page 14, to indicate the different samples used for the representative dot-plots.

A

Adjusted Figure 3A

3. Can the authors define why $\gamma\delta T$ cells would show activation against non-tumorous (NT) tissue in 3C? If it is expected that only tumor-reactive TILs were expanded via the REP protocol, why would there be activation against non-tumor tissue? Could this be a potential therapeutic safety issue? Please address if not already covered in the discussion.

Indeed, $\gamma\delta T$ cells display a level of activation against the NT. A similar effect was observed with $\alpha\beta T$ cells generated from adult lesions against RCC and NSCLC (van Asten et al. Oncoimmunology and de Groot et al. Oncoimmunology), where expanded TILs show a higher response against healthy tissue than when cultured in medium alone. However, in both the pediatric neuroblastoma derived TILs as in the adult derived TILs, the response is highest in the co-culture with tumor cells. We therefore consider the $\gamma\delta T$ TILs as tumor specific. We now mention this point in the discussion (page 9).

Figure 4:

1. Please provide IFN γ /TNF staining for unstimulated $\gamma\delta T$ subsets for comparison to PMA/ionomycin stimulated cells in Figure 4C/D.

Please be advised that the staining for unstimulated $\gamma\delta T$ cell subsets is provided in Figure 4E (now clarified on page 8).

2. Were statistics performed for comparisons provided in 4F? Without any statistical comparisons, it is hard to confirm the findings reported in lines 214-220 and the conclusion that V δ 1 and V δ 3 are the prime $\gamma\delta T$ cell subsets responding to autologous tumor digests. This should also be addressed in the abstract, which without any statistical confirmation, overstates the role of V δ 1 and V δ 3 Ts (Line 36).

We now included statistics in Figure 4F (see below). This clearly shows that only Vd1 and Vd3 T cells respond effectively to tumor digest, and Vd3 cells may respond even better.

F

Adjusted Figure 4F (containing statistics)

Discussion:

1. Lines 222-223 state, "In summary, we here provide first evidence that TIL products can be generated from pediatric neuroblastoma tumor lesions." This statement may not be entirely accurate considering at least one previous literature report that demonstrates TIL expansion from primary neuroblastoma tissues (Hurtado...Anderson et al, 2019). Please adjust accordingly and also include description of how findings of this report substantiate or differ from the previous work. Interestingly, this report also provides description of intratumoral $\gamma\delta$ Ts.

We respectfully disagree. As indicated in the results section (page 4), the study of Hurtado et al. does not use the protocol that is currently used in the clinic for expanding TILs: it uses α CD3 antibodies from day 0 onwards. Adding α CD3 so early drives indeed T cell expansion, but preferentially that of blood contaminants and non-tumor specific non-exhausted T cells. Therefore, the clinically used expansion protocol only adds α CD3 in the second (REP) phase of expansion, i.e. from day 11-14 onwards. Therefore, these protocols cannot be directly compared. We realize that this small but relevant difference of culture conditions was not sufficiently explained, and we have now again highlighted it in the discussion (page 9).

2. One very interesting aspect of the manuscript is the differences between pediatric and adult $\gamma\delta$ T cells and their presence/expansion/activity against neuroblastoma tumors both as TILs and potentially as ACT approaches. It would be nice if the authors could perform additional experiments to further develop this aspect of their story.

We agree. However, because the tumor material from the children is very limited, we are at this stage not in the position to provide such data. We will mention this point in the discussion (page 9). Please be advised that this study is a proof-of-concept study with the prime goal to test whether TIL therapy is a viable option for neuroblastoma patients. We therefore believe that this study already in itself provides important insights.

Points of Minor Revision:

1. In Figure 1, depiction of the gating strategy could be clarified for audiences with limited phenotyping experience. For example, the authors should include clarity that the NK cell gate (A, bottom row, farthest left) originated from the parental CD3⁻ gate (A, top row, farthest right) and that remaining T cell gates (A, bottom row, plots 2-3) originated from parental CD3⁺γδT⁻ gate (A, top row, farthest right). If the assumed gating strategy, described above, was not indeed the gating tree the authors should provide further clarification within the figure.

We appreciate this suggestion, and now added arrows as shown below for clarification.

Adjusted Figure 1A

Adjusted Supplemental Figure 1E

2. While the text is clear regarding the expansion components and timing of pre-REP/REP phases, some readers might appreciate a diagram depicting the expansion protocol used within Figure 2 as visual summary. Additionally, it could be helpful to include a legend of what black (expansion >500-fold) vs. red (expansion <500-fold) dots symbolize within the figure itself.

Again a very good suggestion, we included a diagram to Figure 2 as **NEW Figure 2A** (also see below). As to the red and black dots, we have included such legend in Figure 2.

New figure 2A

3. It is a somewhat confusing that the summary table provided in Supplemental Figure 1 provides mean but the corresponding plots in Figure 2 use median. In the text average is provided - providing some sort of uniformity between the three could make the results easier to follow.

We thank the reviewer for pointing this out. We now adjusted the table in Supplemental Figure 1 to median.

4. Panels and text for Figure 2A,C-F and Supplemental Figure 1D are somewhat hard to follow. Data is repeated (for purpose of different comparisons) but this isn't immediately clear. Consider revising for clarity or condensing into one figure if possible?

To clarify that the data from Figure 2 are plotted again but with a different cut-off, we now added an additional legend in Supplemental Figure 1F (previously Supplemental Figure 1D).

Supplemental Figure 1F

August 13, 2025

RE: Life Science Alliance Manuscript #LSA-2025-03249-TR

Dr. Monika C Wolkers
Sanquin
Plesmanlaan 125
Amsterdam 1066CX
Netherlands

Dear Dr. Wolkers,

Thank you for submitting your revised manuscript entitled "γδ T cells are the prime anti-tumoral T cells in pediatric neuroblastoma". Your revised manuscript was evaluated by two of the original reviewers whose comments are appended below. As you will note, the reviewers have acknowledged your revised manuscript. Reviewer 3 has a minor follow-up point, on γδ TILs as tumor specific, that we encourage you to address. We agree to their overall recommendation and we would be happy to publish your paper in Life Science Alliance, pending final revisions necessary to meet our formatting guidelines.

- Please include a 'Data Availability' statement for any data (for instance FACS data) that is deposited in a public database along with the repository name and persistent identifier (DOI, accession number, or permanent URL). Kindly state the availability of source data in your manuscript.
- Please provide details on approval and consent statement for adult tumour samples described in Figure S2C.
- Please add a Running Title in our system.
- Please move Figure legends/Supplementary Figure legends/Table legends at the end of manuscript file.
- Please use the [10 author names, et al.] format in your references (i.e. limit the author names to the first 10).
- There are callouts for Figure S2D-E in the manuscript text, while these labels are not presented in the figure. Please rectify this discrepancy.
- Please be sure that the authorship listing and order is correct.
- Please add ORCID ID for corresponding author--you should have received instructions on how to do so.
- Please add the X and Bluesky handles of your host institute/organisation as well as your own or/and one of the authors in our system.

A. FINAL FILES:

B. MANUSCRIPT ORGANIZATION AND FORMATTING:

Sincerely,

Sarita Hebbar, PhD
Scientific Editor
Life Science Alliance
<http://www.lsjournal.org>

Reviewer #2 (Comments to the Authors (Required)):

The authors addressed all my questions and I recommend acceptance of the paper.

Reviewer #3 (Comments to the Authors (Required)):

Castenmiller et al. describe a potential novel therapy approach for neuroblastoma, an aggressive pediatric solid tumor, using ex vivo expanded tumor infiltrating lymphocytes (TILs). Evidence is presented that TILs are present in primary neuroblastoma tumors, even following exposure to standard treatment approaches including chemotherapy and radiation (MIBG). The TILs present were expanded ex vivo using IL-2 as stimuli under the REP protocol and produced expanded TIL products containing CD8+ T, conventional (CD4+) T, and $\gamma\delta$ T cells. The authors present proof-of-concept fighting that $\gamma\delta$ T cells expanded within the TIL product contribute anti-tumor activity.

The authors' revisions to both the data presentation and text of the manuscript are appreciated. It is understood that the availability of tumor tissue limits many of the additional studies suggested by all reviewers. In the absence, the authors have worked to highlight the limitations of the study through revised discussion and stress that the current work is proof-of-concept with follow up studies ongoing.

A remaining comment:

Figure 3, #3. Can the authors define why $\gamma\delta$ T cells would show activation against non-tumorous (NT) tissue in 3C? If it is expected that only tumor-reactive TILs were expanded via the REP protocol, why would there be activation against non-tumor tissue? Could this be a potential therapeutic safety issue? Please address if not already covered in the discussion.

Author Response: Indeed, $\gamma\delta$ T cells display a level of activation against the NT. A similar effect was observed with $\alpha\beta$ T cells generated from adult lesions against RCC and NSCLC (van Asten et al. Oncoimmunology and de Groot et al. Oncoimmunology), where expanded TILs show a higher response against healthy tissue than when cultured in medium alone. However, in both the pediatric neuroblastoma derived TILs as in the adult derived TILs, the response is highest in the co-culture with tumor cells. We therefore consider the $\gamma\delta$ TILs as tumor specific. We now mention this point in the discussion (page 9).

Follow-up: Thank you for the additional detail. This figure has no statistical analysis, which might decrease reliability of this assertion and potential safety. Or maybe include that this is a preliminary finding of limited scope that needs to be worked out as the TIL-therapy proceeds forward in expanded preclinical testing against both fibroblasts and other normal tissues.

Reviewer #2 (Comments to the Authors (Required)):

The authors addressed all my questions and I recommend acceptance of the paper.

We thank the reviewer for this comment, and are pleased to hear that we addressed all questions.

Reviewer #3 (Comments to the Authors (Required)):

Castenmiller et al. describe a potential novel therapy approach for neuroblastoma, an aggressive pediatric solid tumor, using ex vivo expanded tumor infiltrating lymphocytes (TILs). Evidence is presented that TILs are present in primary neuroblastoma tumors, even following exposure to standard treatment approaches including chemotherapy and radiation (MIBG). The TILs present were expanded ex vivo using IL-2 as stimuli under the REP protocol and produced expanded TIL products containing CD8+ T, conventional (CD4+) T, and $\gamma\delta$ T cells. The authors present proof-of-concept fighting that $\gamma\delta$ T cells expanded within the TIL product contribute anti-tumor activity.

The authors' revisions to both the data presentation and text of the manuscript are appreciated. It is understood that the availability of tumor tissue limits many of the additional studies suggested by all reviewers. In the absence, the authors have worked to highlight the limitations of the study through revised discussion and stress that the current work is proof-of-concept with follow up studies ongoing.

A remaining comment:

Figure 3, #3. Can the authors define why $\gamma\delta$ T cells would show activation against non-tumorous (NT) tissue in 3C? If it is expected that only tumor-reactive TILs were expanded via the REP protocol, why would there be activation against non-tumor tissue? Could this be a potential therapeutic safety issue? Please address if not already covered in the discussion.

Author Response: Indeed, $\gamma\delta$ T cells display a level of activation against the NT. A similar effect was observed with $\alpha\beta$ T cells generated from adult lesions against RCC and NSCLC (van Asten et al. Oncoimmunology and de Groot et al. Oncoimmunology), where expanded TILs show a higher response against healthy tissue than when cultured in medium alone. However, in both the pediatric neuroblastoma derived TILs as in the adult derived TILs, the response is highest in the co-culture with tumor cells. We therefore consider the $\gamma\delta$ TILs as tumor specific. We now mention this point in the discussion (page 9).

Follow-up: Thank you for the additional detail. This figure has no statistical analysis, which might decrease reliability of this assertion and potential safety. Or maybe include that this is a preliminary finding of limited scope that needs to be worked out as the TIL-therapy proceeds forward in expanded preclinical testing against both fibroblasts and other normal tissues.

We thank the reviewer for this follow-up question. Due to the spread in response in the cohort, and the small group size in Figure 3C (n=3), we previously refrained from statistical analysis. However, we now also provide the data normalized to medium control as NEW Figure 3D, and we included statistical analysis. We mention this finding in the results section on page 6.

NEW figure 3D

August 21, 2025

RE: Life Science Alliance Manuscript #LSA-2025-03249-TRR

Dr. Monika C Wolkers
Sanquin
Plesmanlaan 125
Amsterdam 1066CX
Netherlands

Dear Dr. Wolkers,

Thank you for submitting your Research Article entitled " $\gamma\delta$ T cells are the prime anti-tumoral T cells in pediatric neuroblastoma". It is a pleasure to let you know that your manuscript is now accepted for publication in Life Science Alliance. Congratulations on this interesting work.

DISTRIBUTION OF MATERIALS:

Again, congratulations on a very nice paper. I hope you found the review process to be constructive and are pleased with how the manuscript was handled editorially. We look forward to future exciting submissions from your lab.

Sincerely,

Sarita Hebbar, PhD
Scientific Editor
Life Science Alliance
<http://www.lsajournal.org>